# A Novel Form of Arginine-Chitosan as Nanoparticles Efficient for siRNA Delivery into Mouse Leukemia Cells

**DOI:** 10.3390/ijms24021040

**Published:** 2023-01-05

**Authors:** Jixian Luo, Jiangfeng Chen, Yan Liu, Yongji He, Wenjuan Dong

**Affiliations:** 1School of Life Science, Shanxi University, 92 Wucheng Road, Taiyuan 030006, China; 2Institute of Functional Food of Shanxi, Shanxi Agricultural University, 79 Longcheng Street, Taiyuan 030031, China; 3Institute of Environmental Science, Shanxi University, 92 Wucheng Road, Taiyuan 030006, China

**Keywords:** chitosan, arginine, modification, siRNA, transfection

## Abstract

The modification of chitosan (CS) has greatly expanded its application in the field of medicine. In this study, low-molecular-weight chitosan was modified with arginine (Arg) by a simple method. The identification by the Fourier transform infrared spectra (FTIR) showed that Arg was successfully covalently attached to the CS. Interestingly, Arg-CS was identified as nanoparticles by atomic force microscopy (AFM) and transmission electron microscopy (TEM), whose particle size was 75.76 ± 12.07 nm based on Dynamic Light Scattering (DLS) characterization. Then, whether the prepared Arg-CS nanoparticles could encapsulate and deliver siRNA safely was investigated. Arg-CS was found to be able to encapsulate siRNAs in vitro via electrostatic interaction with siRNA; the Arg-CS/siRNA complex was safe for L1210 leukemia cells. Therefore, modification of chitosan by Arg produces novel nanoparticles to deliver siRNA into leukemia cells. This is the first time to identify Arg-CS as nanoparticles and explore their ability to deliver *Rhoa* siRNA into T-cell acute lymphoblastic leukemia (T-ALL) cells to advance therapies targeting Rhoa in the future.

## 1. Introduction

Small interfering RNA (siRNA), a 21–25 bp double-stranded RNA, can effectively silence target genes through specific binding to the mRNA of target genes [1]. However, siRNA application in the treatment of clinical diseases has been restricted because of the transmission barrier and its instability [2,3].

A key challenge to realizing the broad potential of siRNA-based therapeutics is the need for safe and effective delivery methods [4]. siRNA delivery systems can be either viral or non-viral. Viral vectors, with ideal delivery efficiency, have security problems such as mutations [5]. Non-viral vectors, such as liposomes, polymers, and polypeptides, have been developed [6,7,8]. With the development of nanotechnology, nanocarriers including self-assembled nanoparticles, polymers, nanocapsules, and two-dimensional (2D) nanomaterials have been demonstrated to significantly improve siRNA transfection into solid cancer cells, such as human breast cancer cells [9,10], pancreatic cancer cells [11], prostate cancer cells [12], and brain cancer cells [13]. For solid tumors, although there are pathophysiological barriers, including vasculature or non-vasculature barriers, siRNA-based medicine has obvious advantages in terms of stability, drug loading efficiency, pharmacokinetics, targeting ability, safety, and multi-functionality [14]. Unfortunately, such RNAi-based therapies in leukemia have lagged behind their applications in the treatment of solid tumors. Studies in leukemia cells have relied on electroporation or viral vectors, which cannot be translated to the clinical setting [15]. Delivery of RNA into lymphocytic leukemia cells remains a formidable challenge and a bottleneck for developing targeted therapies for this disease [16].

Cationic polymers are most relevant for siRNA delivery due to their ability to interact with oligonucleotides. As a unique polysaccharide, the primary amino group on the chitosan (CS) chain makes it positively charged in the solution [17]. However, CS has promising properties, including high biocompatibility, excellent biodegradability, low toxicity, as well as abundant availability with low production costs [18]. Drug delivery with the help of chitosan-derived nanomaterials has been explored, owing to their inherent ability to deliver both hydrophilic and hydrophobic drug molecules, increase stability, decrease toxicity, and enhance commonly formulated medications. CS-based nanoparticles have been used in glioblastoma tumor therapy [19,20,21]. Therefore, in the present work, Arg-modified CS was prepared and identified, and then its biosafety and probability of delivering siRNAs to mouse T-cell acute lymphoblastic leukemia (L1210) were investigated. We aimed to elucidate the barriers to efficient delivery of siRNA into T-ALL cells to advance therapies in the future.

## 2. Results

### 2.1. Modification of Chitosan by Arginine Using EDC/NHS Produced a Kind of Nanoparticle Material

The surface functional groups of Arg-CS were analyzed by FT-IR spectroscopy and compared with CS. As shown in Figure 1, the FT-IR spectrum of Arg-CS shows a broad absorption band at 3000–3600 cm^−1^ which is attributed to the stretching vibration of N-H or O-H [22,23]. The peak at 2869 cm^−1^ is assigned to the stretching vibration of C-H bonds. The absorption bands at 1150 and 1025 cm^−1^ are ascribed to the asymmetric stretching and skeletal stretching vibration of C-O-C [24,25]. Obviously, these FT-IR characteristic absorption peaks of Arg-CS are very similar to those of CS, suggesting that the combination of Arg and CS does not destroy the skeleton structure of CS. However, compared to CS, three bands at 1670, 1540, and 1290 cm^−1^ occurred in the spectrum of Arg-CS, which correspond to the I, II, and III bands of a typical amide group [26,27]. Since pure CS or Arg does not contain the amide group, it can be speculated that the carboxyl group of Arg reacted with the primary amine group of CS. This result demonstrates that Arg has been successfully bound to CS by an amide bond using the EDC/NHS method.

Then, the surface morphology and size distribution of the prepared Arg-CS were observed by AFM. The three-dimensional image of AFM (Figure 2A) and the two-dimensional image (Figure 2B) showed that the Arg-CS product was nanoparticle; the height of the Arg-CS sample on the line in Figure 2B is 2, 6, and 8 nm (Figure 2C). Arg-CS are generally distributed in the range from 5.00 to 15.00 nm in height, with the major height about 13.00 nm (Figure 2D). The TEM image of Arg-CS further verified that the prepared Arg-CS is a type of nanoparticle (Figure 2E).

### 2.2. The Prepared Arginine-Chitosan Nanoparticles Encapsulate siRNA In Vitro via Electrostatic Interaction

The possibility of Arg-CS to complex with siRNA was investigated through gel retardation experiments. The results showed that under an external electric field of 100 V, the band corresponding to naked siRNAs were present at the corresponding base pair position in lane 2; when the weight ratio of Arg-CS:siRNA was 1:1, major siRNA was retardated in the gel well by Arg-CS; and when the weight ratios of Arg-CS:siRNA were greater than or equal to 10:1 (10:1, 20:1, 50:1, and 100:1), the motivation of siRNA were almost completely blocked by Arg-CS (Figure 3A). Similarly, siRNA retardation was also present in CS dissolved in NaAc (Figure 3B). This result showed that Arg modification did not alter the property of CS to complex with siRNA.

To investigate the physicochemical interaction between Arg-CS and siRNA, Arg-CS/siRNA were prepared, and a series of characterizations of the assembly morphology were subsequently performed. Pure siRNA displayed negative surface charge (−1.67 ± 0.17 mV), while a complex with positively charged Arg-CS recorded that the zeta potential increased from 0.57 ± 0.14 to 16.63 ± 1.18 mV. Similar interactions were present at the CS/siRNA complex, suggesting that Arg-CS and siRNA interact with each other via electrostatic interaction (Table 1). The DLS test results showed that the particle size distribution of the Arg-CS/siRNA complex was from 124.83 ± 11.58 to 1261.90 ± 438.88 nm.

### 2.3. The Arg-CS/Rhoa siRNA Complexes Are Safe for L1210 Cells

Before investigating the possibility to deliver siRNA by Arg-CS nanoparticles, the effects on L1210 cell viability were investigated by MTT assay. The results showed that Arg-CS/siRNA were safe for L1210 cells within the range. CS/*Rhoa* siRNA complexes at the ratios of 20:1 and 50:1 were safe for L1210 cells (Figure 4).

### 2.4. Arginine-Chitosan Nanoparticles Can Deliver siRNA to L1210 Cells

Based on the results of the gel retardation experiment and the effect of Arg-CS on cell viability, different weight ratios were used to investigate the possibility of using Arg-CS to deliver *Rhoa* siRNA to L1210 cells. It was shown that the green, fluorescent signal, which represents FAM-labeled *Rhoa* siRNAs, was present in Arg-CS-transfected L1210 cells, while little or none was present in cells transfected by CS. The transfection efficiency at 20:1 was higher compared with those transfected at other ratios or by CS (Figure 5). These results indicate that Arg-CS nanoparticles is an efficient nanomaterial for delivering siRNAs to mouse leukemia L1210 cells.

### 2.5. Rhoa Is Knocked Down in L1210 Cells Transfected by Arg-CS/Rhoa siRNA

To analyze the knockdown efficiency of *Rhoa* siRNA delivered by Arg-CS, L1210 cells were transfected, and a Western blot assay was conducted. The results showed that Rhoa was knocked down in the Arg-CS/*Rhoa* siRNA-transfected L1210 cells compared to CS/*Rhoa* siRNA. This result demonstrated the advantage of Arg-CS to efficiently deliver *Rhoa* siRNA (Figure 6).

## 3. Discussion

Lymphocytes are implicated in many diseases, such as hematologic malignancies, inflammation, autoimmunity, transplant rejection, and viral infections [15,28]. Safe delivery of siRNA into lymphocytes holds great promise and opens novel therapeutic possibilities [29]. We previously showed that RhoA/RhoC siRNAs inhibit T-ALL cell migration mediated by SDF-1/CXCR4 in vitro [30]. Therefore, appropriate systems that could deliver RhoA/RhoC siRNAs to leukemia cells in a safe manner are significant for clinical application in leukemia therapy. In the present work, the ability of Arg-CS to safely deliver *Rhoa* siRNA into mouse leukemia cells is precious for a leukemia cure.

The solubility and cations of CS have been shown to be changed by acylation, N-alkylation, chelation, oxidation, oxygen-containing inorganic acid esterification, cross-linking, and graft polymerization reactions [31]. For example, the dispersibility and cations of CS are usually enhanced by the reaction with acid chlorides and acid anhydrides, forming amides, or esters [32]. Theoretically, ester bonds and amide bonds can be formed between the carboxyl groups of amino acids and the 2-amino, 3-hydroxy, or 6-hydroxy groups of the CS molecule [33]. In the present work, Arg-CS was successfully prepared using EDC and NHS as cross-linking agents (Figure 1). Predictably, the cations and biocompatibility of CS can be enhanced by the reaction between the 2-amino, 3-hydroxy, or 6-hydroxy group of CS and the carboxyl group of Arg, forming amide bonds or ester bonds [34,35].

Among the nanoparticle-based RNAi delivery strategies utilized to deliver RNAi molecules into lymphocytes, a polymer-based RNAi delivery system often utilizes cationic polymers [36]. CS is considered to be the most important natural polymer gene carrier [37]. Under acidic conditions, a stable nanocomplex can be formed by the electrostatic interaction between positively charged CS molecules and negatively charged siRNAs [38]. In this case, siRNA is not easily degraded by nucleases in vivo under the protection of CS. The high positive charge density at reduced pH may enable endosomal escape [39]. A CS-siRNA nanodrug can be delivered to cells and cause gene silencing [40]. Interestingly, we detected the particle size and characteristics of Arg-CS and found that the prepared Arg-CS itself was in a nanoparticle form (Figure 2), providing rationality for Arg-CS to deliver siRNA. Consistently, we found that Arg-CS and siRNA formed a nanocomplex with siRNA via electrostatic interaction (Table 1 and Figure 3).

CS is degradable, so it has minimal side effects on the cell. The biosafety of Arg-CS/siRNAs as well as siRNA delivery capability into mouse leukemia cells were investigated further. The results showed that Arg-CS/siRNAs are safe for L1210 cells (Figure 4) and that Arg-CS nanoparticles compete with CS in their ability to deliver siRNAs to L1210 cells (Figure 5 and Figure 6). To enhance the cations and thereby enhance the absorption of CS into DNA or RNA, amino acids whose R groups contain a positive charge were preferred. The R group of Arg is a positively charged guanidine group [41], and Arg modification enhances the positive charge of CS. The first reason for Arg-CS to compete with CS in siRNA delivery may be that the Arg modification enhances the electrostatic interaction. Arg contains a large number of hydrogen and nitrogen atoms in its R group, which form hydrogen bonds with the substrate [42,43]. The internalization mechanism of the Arg-CS/siRNA complex is possibly due to the guanidine groups of Arg that form hydrogen bonds with polyhydroxy compounds in cell membranes that facilitate the translocation across cell membranes [44]. In addition, the guanidine groups of arginine also have a strong affinity to heparan sulfate, which assists with adhering to the cell membrane [45]. The second reason for Arg-CS’s in successfully delivering siRNAs may be due to the properties of nanoparticles. Nanomaterials have a smaller volume and a larger specific surface area compared with non-nanomaterials, which can bind more siRNA and prevent the target cell’s lipid bilayers from severe damage [46]. Considering its low toxicity to L1210 leukemic cells and high efficiency for siRNA delivering, Arg-CS nanoparticles may be effective transfection reagents for L1210 cells or a candidate material for a leukemia target treatment.

## 4. Materials and Methods

### 4.1. Main Reagents

FAM-labeled *Rhoa* siRNA was synthesized by Sangon Biotech (Shanghai) Co., Ltd. (Shanghai, China). The sequences of negative control (NC) and *Rhoa* siRNA were as follows:

*Rhoa* siRNA sense: UGGCGGAUAUCGAGGUGGAUGGGAATT

*Rhoa* siRNA antisense: UUCCCAUCCACCUCGAUAUCCGCCATT

NC sense: UUCUCCGAACGUGUCACGUTT

NC antisense: ACGUGACACGUUCGGAGAATT

Chitosan (CS) was purchased from Solarbio, and its molecular weight (480~776 Da) was measured by matrix-assisted laser desorption time-of-flight mass spectrometry. 1-(3-Dimethylaminopropyl)-3-ethylcarbodiimide (EDC), N-hydroxysuccinimide (NHS), arginine (Arg), hydroxylamine, sodium acetate, dimethyl sulfoxide, polyethylene glycol (PEG) 20,000, and deuterated heavy water were from Solarbio. NaAc sodium hydroxide, was from Macklin. Lipofectamine 2000 was from Invitrogen. Monoclonal antibodies to Rhoa and actin were from Cell Signaling Technology and Sangon Biotech (Shanghai, China) Co., Ltd., respectively.

### 4.2. Synthesis and Characterization of Arginine-Chitosan

#### 4.2.1. Synthesis of Arginine-Chitosan (Arg-CS)

The low-molecular-weight chitosan CS was modified by Arg according to the previous description with a slight modification [47]. An amount of 1.00 g of CS was completely dissolved in 100 mL of CH3COOH, followed by continuous stirring at 1000 rpm/min. Then, 1.00 g of NHS and 1.80 g of EDC were added to the CS solution under constant stirring. After 0.5 h of reaction, 2.00 g of Arg was added under constant stirring. After continuous stirring for 24 h, the above reaction was terminated by adding 0.50 g of H3NO under constant stirring. A 8–14 kDa dialysis membrane was used for dialysis of the crude product for 5 days, and then PEG 20,000 was used outside of the dialysis membrane for 24 h to absorb water and concentrate the dialysis product. The product was pre-frozen in a −20 °C refrigerator for 0.5 h and freeze-dried in a vacuum freeze dryer for 24 h. Finally, the Arg-CS powder was collected for further use.

#### 4.2.2. Characterization of Arginine-Chitosan

The structure of Arg-CS was characterized by a Thermo Scientific Nicolet iS50 FT-IR spectrometer (Thermo Fisher Scientific, Waltham, MA, USA). The morphology and particle size of Arg-CS were characterized by atomic force microscopy (AFM) and transmission electron microscopy (TEM).

### 4.3. Preparation and Characterization of the Arg-CS/siRNA Complex or CS/siRNA Complex

Arg-CS or CS was dissolved in NaAc (pH = 5.5) at different concentrations, including 2, 1, 0.4, 0.2, and 0.02 μg/μL. A 5 μL sample of each concentration and 5 μL of siRNA (0.02 μg/μL) in DEPC water were added into RNAase-free Eppendorf tubes to make the CS-Arg/siRNA complex sample at different weight ratios (1:1, 10:1, 20:1, 50:1, and 100:1), mixed in a vortex machine for 30 s, and then incubated at room temperature for 30 min to prepare test complexes.

In the agarose gel electrophoresis retardation experiment, 10 μL of prepared complexes were used to run RNA agarose gel electrophoresis at 100 V for 30 min, and then the gel was scanned under FluorChem HD2 Alpha (ProteinSimple, Shanghai, China).

A ZETA potential analyzer was used to measure the nanoparticle size of a 10 μL prepared Arg-CS/siRNA complex or CS/siRNA complex.

### 4.4. Cell Culture

A mouse T-cell acute lymphoblastic leukemia cell line L1210 was kindly provided by the former Institute of Cell Biology of the Chinese Academy of Sciences (Shanghai, China) and maintained at 37 °C with 5% CO_2_ in Dulbecco’s Modified Eagle’s Medium (DMEM), containing 10% fetal bovine serum, 100 U/mL of penicillin, and 100 µg/mL of streptomycin.

### 4.5. Cell Viability Analysis

L1210 cells were counted and seeded in 96-well plates at a cell concentration of 1000/well. After incubation at 37 °C for 24 h in a 5% CO_2_ incubator, Arg-CS/siRNA complex or CS/siRNA complex was added to L1210 cells (*n* = 3). A volume equal to that of NaAc was used as the control. After incubation at 37 °C for 4 h, 10 μL of MTT (5 μg/μL) was added to each well. After incubation at 37 °C for another 4 h, cells were washed with PBS. An amount of 150 μL of DMSO was added to each well, then mixed and incubated for 10 min. The absorbance of the L1210 cells was measured at 570 nm using a microplate reader. Then the cell viability was calculated by dividing the absorbance number of each group by that of the control group.

### 4.6. Transfection

A total of 2 × 10^5^ L1210 cells in the logarithmic growth phase were seeded into each well of a 6-well cell culture plate. After incubation for 24 h at 37 °C, the medium was replaced with DMEM without serum before transfection. An Arg-CS/siRNA complex or CS/siRNA complex was prepared as described above, diluted with 500 µL of DMEM, and added to each well containing L1210 cells. After 4 h of culture at 37 °C in a 5% CO_2_ incubator, cells were collected, washed with PBS 3 times, and then resuspended in PBS, and pictures were taken randomly under the twenty-fold objective of a fluorescence microscope (Olympus, Tokyo, Japan). In the experiment to detect the knockdown efficiency of *Rhoa* siRNA, 4 μg/μL of Arg-CS or CS was complexed with 20 μL of siRNA (0.2 μg/μL) at a 20:1 weight ratio as described above. Four hours after transfection, the cells were washed with PBS and cultured for another 48 h in DMEM medium plus 10% FBS. Then cells were collected, and a Western blot assay was conducted using Rhoa antibodies, or actin antibodies.

### 4.7. Data Analysis

The data obtained was analyzed using GraphPad Prim 8.0 software, and the experimental results were expressed as mean ± standard deviation. The statistical difference between the two groups of data was analyzed by the Mann–Whitney nonparametric *t* test, * *p* < 0.05 indicates a significant difference and ** *p* < 0.01 indicates an extremely significant difference.

## 5. Conclusions

In the present article, the Arg-CS material was successfully prepared using EDC and NHS. The morphology and particle size characterization of the prepared Arg-CS material were conducted by AFM and TEM, which showed that it is a nanoparticle material with a particle size 75.76 ± 12.07 nm. Arg-CS nanoparticles have the ability to encapsulate siRNA and safely deliver *Rhoa* siRNA to mouse leukemia L1210 cells. Therefore, modification of chitosan by Arg produces novel nanoparticles to deliver *Rhoa* siRNA into leukemia cells.

## Figures and Tables

**Figure 1 ijms-24-01040-f001:**
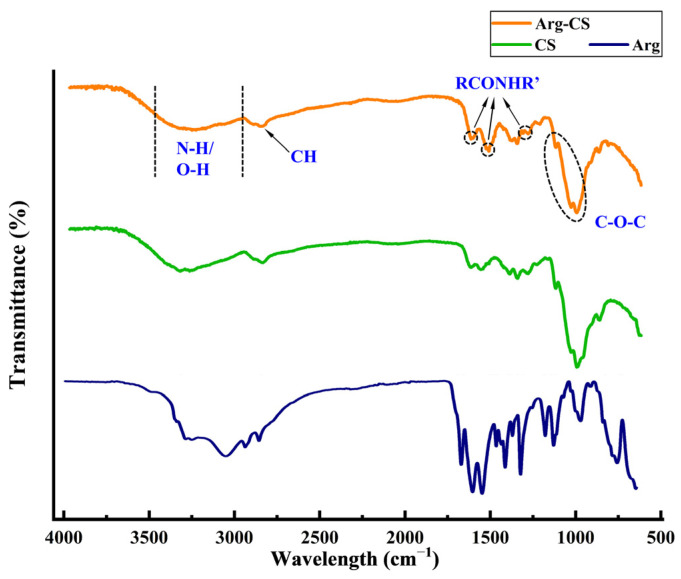
Amide group was formed between Arg and CS in the prepared Arg-CS. The FT-IR spectra of Arg-CS (orange line), Arg (blue line), and CS (green line). CS, or prepared Arg-CS, was detected by a Thermo Scientific Nicolet iS50 FT-IR spectrometer; the characteristic peaks were assigned arrows.

**Figure 2 ijms-24-01040-f002:**
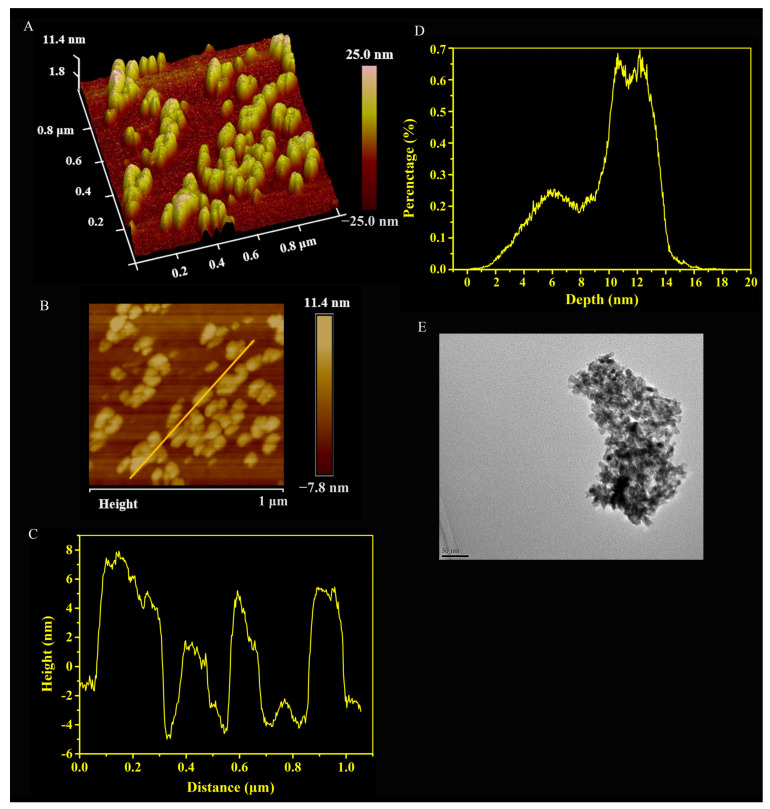
Characterization of Arg-CS by atomic force microscopy and transmission electron microscopy. (**A**) the 3D image of AFM, (**B**) the two-dimensional image of AFM, (**C**) the height distribution of Arg-CS along the line in (**B**), and (**D**) the height distribution of all the Arg-CS shown in (**B**). The representative TEM image of Arg-CS (**E**).

**Figure 3 ijms-24-01040-f003:**
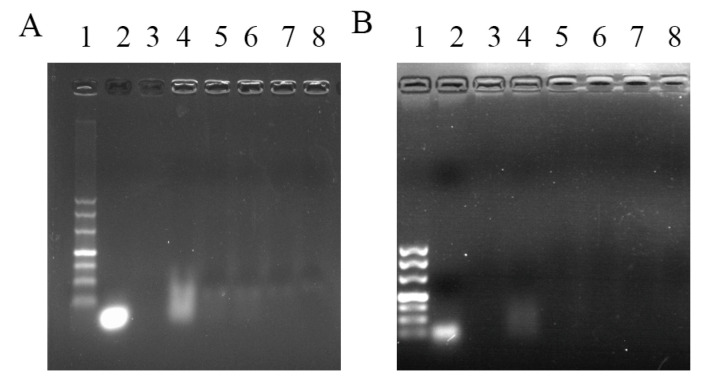
siRNA was encapsulated by Arg-CS and retarded by electrophoresis in agarose. (**A**) Arg-CS and (**B**) CS were used to encapsulate siRNAs. Lane 1: DNA marker (from top to bottom of the lane: 500 bp, 400 bp, 300 bp, 200 bp, 150 bp, 100 bp, and 50 bp); lane 2: siRNA; lane 3: Arg-CS or CS; lanes 4–8: weight ratios of Arg-CS or CS to siRNA equal to 1:1, 10:1, 20:1, 50:1, and 100:1, respectively. The concentrations of Arg-CS and siRNA were 0.02 μg/μL. The concentration of NaAc was 0.1 M (pH = 5.5).

**Figure 4 ijms-24-01040-f004:**
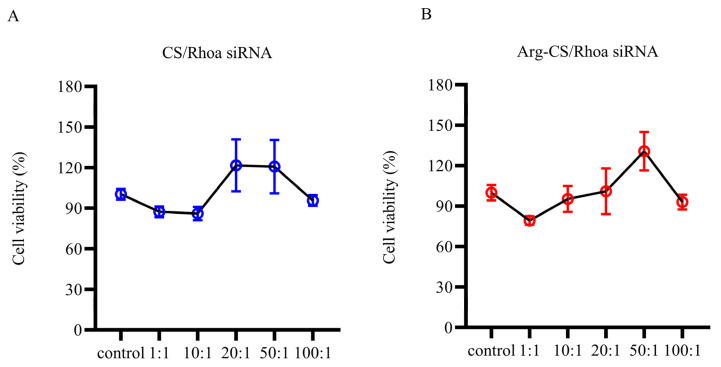
The cell viability of L1210 cells was not changed by Arg-CS/siRNA. (**A**) CS/siRNA and (**B**) Arg-CS/siRNA complexes were prepared as described in the Section 4, and cell viability was measured by the MTT assay (*n* = 3).

**Figure 5 ijms-24-01040-f005:**
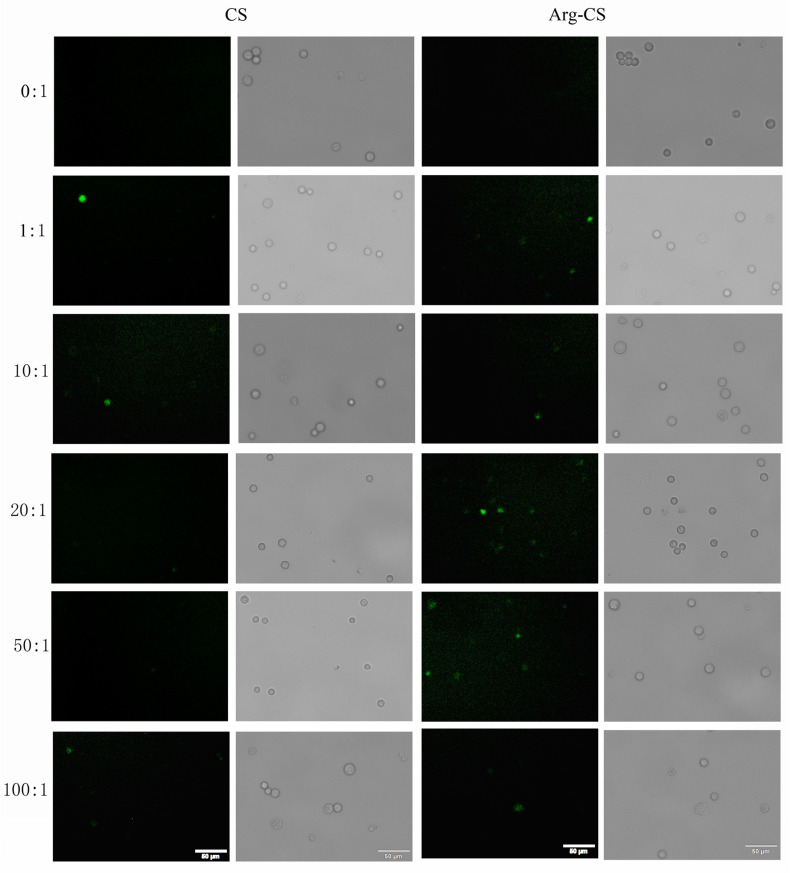
*Rhoa* siRNAs could be transfected by Arg-CS. L1210 cells were transfected as described in Section 4 at different ratios, and cells were observed and photographed under a fluorescence microscope 4 h after transfection; typical images are shown. Green flurescent cells stand for the cells into which FAM-*Rhoa* siRNAs have been succesfully transfected. Circles stand for all cells in each image.

**Figure 6 ijms-24-01040-f006:**
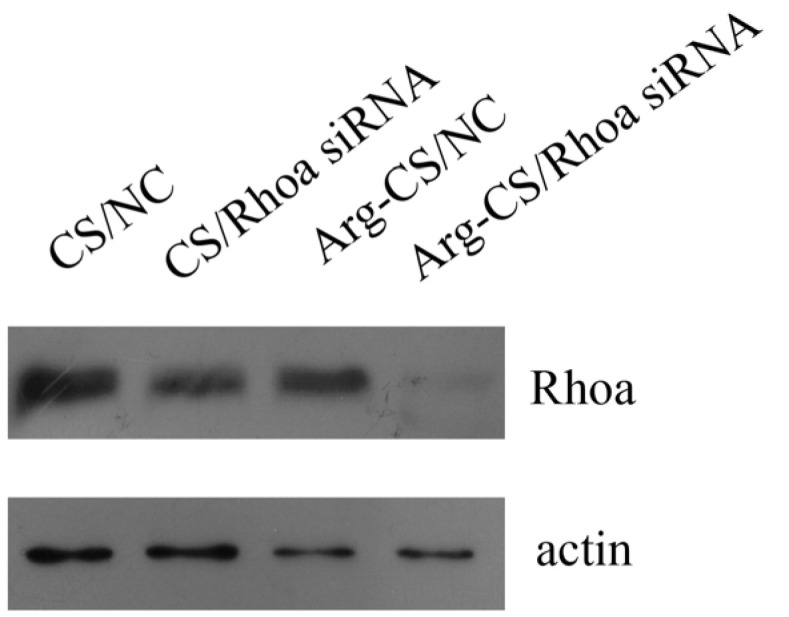
Rhoa could be knocked down in L1210 cells transfected with a 20:1 nanoparticle/siRNA ratio, as described in Section 4. Cells were collected, and antibodies to Rhoa and actin were used in a Western blot assay. Actin was used as a loading control.

**Table 1 ijms-24-01040-t001:** The summary of physiochemical properties of the Arg-CS/siRNA complex.

Sample Name	Mass Ratio (m/m)	DSL (nm)	PDI	Zeta Potential (mV)
siRNA	—	450.87 ± 53.67	0.34 ± 0.17	−1.67 ± 0.17
CS/siRNA	1:0	483.13 ± 82.34	0.33 ± 0.31	30.95 ± 2.33
	1:1	273.17 ± 26.32	0.62 ± 0.11	1.74 ± 0.95
	10:1	1299.13 ± 450.22	0.73 ± 0.21	17.67 ± 2.66
	20:1	1673.13 ± 261.43	1.10 ± 0.36	17.01 ± 3.93
	50:1	1865.80 ± 99.50	1.75 ± 0.18	21.5 ± 0.95
	100:1	1122.23 ± 277.23	0.66 ± 0.32	20.01 ± 6.05
Arg-CS/siRNA	1:0	75.76 ± 12.07	0.43 ± 0.11	23.24 ± 1.91
	1:1	124.83 ± 11.58	0.60 ± 0.39	0.57 ± 0.14
	10:1	453.87 ± 100.75	0.63 ± 0.12	2.23 ± 0.08
	20:1	440.30 ± 178.21	0.73 ± 0.07	8.62 ± 0.53
	50:1	1261.90 ± 438.88	0.68 ± 0.36	9.2 ± 0.64
	100:1	296.83 ± 5.13	0.65 ± 0.11	16.63 ± 1.18

Arg-CS or CS in NaAc (0.1 mol/L, pH = 5.5) and siRNAs in DEPC water were mixed at the weight ratios of 1:1, 10:1, 20:1, 50:1, and 100:1 to form an Arg-CS/siRNA or CS/siRNA complex. The zeta potentials, DSL and PDI, were measured by nanoparticle size and a ZETA potential analyzer.

## Data Availability

The data presented in this study are available in the article.

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
