# Peer review of "A Novel Form of Arginine-Chitosan as Nanoparticles Efficient for siRNA Delivery into Mouse Leukemia Cells"

_ijms, 2023, doi:10.3390/ijms24021040_

Round 1
Reviewer 1 Report
1. No knockdown efficiency was provided. This result is needed for publication in Int. J. Mol. Sci.
2. Transfection experiments provided in this manuscript is just cellular uptake. Please discuss more precisely.
3. what is the shape of unmodified CS? Please study. The following experiments of siRNA complexation, cell toxicity, cellular uptake, and knockdown efficiency should use unmodified CS as a control.
4. In the introduction, the authors should emphasize the advantage of nanotechnology to advance the function (https://doi.org/10.1021/jacs.0c09029).
Author Response
- No knockdown efficiency was provided. This result is needed for publication in Int. J. Mol. Sci.
Response: Thank you for your suggestion, the transfection experiment showed that Arg-CS nanorods have the ability to deliver siRNA into leukemic cells (Figure 5 and Figure 6). This manuscript focus on the novel found that Arg-CS is a kind of nanomaterial that have the potential to safely deliver siRNA into leukemia cells. We focus on the siRNA delivering step, since a key challenge to realizing the broad potential of siRNA-based therapeutics is the need for safe and effective delivery methods.
- Transfection experiments provided in this manuscript is just cellular uptake. Please discuss more precisely.
Response: We really appreciate your valuable suggestion, we specified it as cellular uptake in the manuscript.
- what is the shape of unmodified CS? Please study. The following experiments of siRNA complexation, cell toxicity, cellular uptake, and knockdown efficiency should use unmodified CS as a control.
Response: As suggested by the reference, CS is a kind of macromolecule paper [International Journal of Biological Macromolecules 89 (2016) 518–526]. We observed the shape of unmodified CS by SEM. The result showed that it's not a kind of nano-material (Figure S1). Since our work is based on the finding that Arg-CS is a kind of nanorods and the literature showing that nanomaterials have advantages in siRNA delivery, we focused on Arg-CS instead of both Arg-CS and CS. In the transfection experiment, Arg-CS have advantages in siRNA delivery comparing with CS (Figure 5 and Figure 6).
Figure S1. SEM result of CS
- In the introduction, the authors should emphasize the advantage of nanotechnology to advance the function (https://doi.org/10.1021/jacs.0c09029).
Response: Following your suggestion, we have written a paragraph in the manuscript.
Reviewer 2 Report
The paper submitted by Luo et al. investigates the preparation of some CS-Arg/siRNA complex and their delivery into mouse leukemia cells.
1. The introduction section is very poor and the references related to other types of systems for the delivery of siRNA must be detailed in order to highlight the originality of the proposed strategy.
2. The advantage of modifying the CS with Arg is not visible. In this context, the authors must compare the results obtained for CS-Arg/siRNA complex with those for CS/siRNA.
3. the titles of the subsections 2.1; 2.2; 2.3 and 2.4 must be modified. There should be a general title, not a conclusion.
4. add the FTIR spectra of Arg and for the CS-Arg/siRNA complex
5. AFM analysis of the CS-Arg/siRNA complex. After the formation of the complex, the morphology is the same?
6. cell viabilities of CS-Arg/siRNA must be provided.
7. the procedure for the preparation of the CS-Arg/siRNA complex is not provided.
8. line 229: why PEG was use for concentrate? which kind of PEG?
Author Response
- The introduction section is very poor and the references related to other types of systems for the delivery of siRNA must be detailed in order to highlight the originality of the proposed strategy.
Response: We have rewritten the introduction following your suggestion.
- The advantage of modifying the CS with Arg is not visible. In this context, the authors must compare the results obtained for CS-Arg/siRNA complex with those for CS/siRNA.
Response: We have compared the results obtained for CS-Arg/siRNA with those for CS-siRNA in the revised manuscript (Figure 6 and Figure 7) . The result showed that Arg modified CS has advantage in siRNA deliver efficiency.
- the titles of the subsections 2.1; 2.2; 2.3 and 2.4 must be modified. There should be a general title, not a conclusion.
Response: Following your suggestion, we have revised the titles of the subsections 2.1; 2.2; 2.3 and 2.4
- add the FTIR spectra of Arg and for the CS-Arg/siRNA complex
Response: We have added the FTIR spectra of Arg in the revised figure 1 as a control. This experiment was to analyze the surface functional groups of Arg-CS. Pure CS or Arg does not contain the amide group, which was present in the Arg-CS sample, suggesting that the carboxyl group of Arg have covalently bound to the amine group of CS. The purpose of this experiment was to detect whether CS has been modified by Arg successfully. At the present, Arg-CS/siRNA complex hasn't been introduced. Therefore, CS-Arg/siRNA was not relevant for figure 1. The interaction between Arg-CS and siRNA were demonstrated to be electrostatic interaction in another result (Table 1 ). Furthermore, each experiment for FTIR needs 0.01g sample, corresponding to 300 OD. Usually the amount of siRNA commercially synthesized was 1-5 OD. Therefore, it's a pity that we were not able to provide adequate siRNA for FTIR.
- AFM analysis of the CS-Arg/siRNA complex. After the formation of the complex, the morphology is the same?
Response: After the formation of the complex, ArgCS/siRNA complex was larger than ArgCS (Table 1).
- cell viabilities of CS-Arg/siRNA must be provided.
Response: cell viabilities of CS-Arg/siRNA orCS/siRNA were provided in Figure 5 in the revised manuscript
- the procedure for the preparation of the CS-Arg/siRNA complex is not provided.
Response: In the materials and methods section, the procedure has been provided in the subtitle of 4.3 in the revised manuscript.
- line 229: why PEG was use for concentrate? which kind of PEG?
Response: PEG20000 was broadly used in protein extraction to absorb water. Following your suggestion, its use was specifically referred in 4.1 and 4.2.1 in the revised manuscript.
Reviewer 3 Report
The manuscript written by Luo et al. is about preparing a non-viral vehicle made up of arginine and chitosan to be used for the transport of two siRNA oligonucleotides. This manuscript involves the synthesis, characterization, and cell culture experiments. The manuscript is interesting and well-organized. However, this manuscript is not appropriate to be published in Int. J. Mol. Sci. To increase the manuscript’s quality, the authors should include additional experiments to confirm the potential of the siRNA complexes in vitro.
(1) The authors should study the release of siRNA from the complexes by native polyacrylamide gel electrophoresis using heparin.
(2) Although the authors have shown AFM results, they should also include DLS and PDI values of the complexes
(3) Why did the authors use just one ratio to calculate the zeta-potential? It would be interesting to study the same ratios the authors used in the electrophoresis with agarose. In this sense, one might characterize how the surface charge is changing as the ratios tend to increase.
(4) It is not necessary to include the zeta-potential chromatograms in the figure. The values should be listed in a table together with the DLS and PDI.
(5) Figure 7. The authors should study other ratios in order to evaluate the transfection efficiency.
(6) The authors have used two siRNA oligonucleotides (RhoA and RhoC) however the silencing of the complexes using RNA interference is missing in this manuscript.
(7) Materials and methods; lines 213-214. What does NC stand for? I think these siRNA oligonucleotides have not been used in the manuscript.
(8) Line 221. Please, include the number of mmol of the reagents used for the preparation of the material
(9) Line 263. The authors should indicate in the experimental part whether siRNA complexes were transfected using serum-free conditions

Author Response
- The authors should study the release of siRNA from the complexes by native polyacrylamide gel electrophoresis using heparin.
Response: Thank you for your suggestion, siRNA was used in this manuscript according to reference [Biomaterials 31 (2010) 8780e8788].
- Although the authors have shown AFM results, they should also include DLS and PDI values of the complexes.
Response: Following your suggetion, DLS and PDI values have been added in Table 1 in the revised manuscript.
- Why did the authors use just one ratio to calculate the zeta-potential? It would be interesting to study the same ratios the authors used in the electrophoresis with agarose. In this sense, one might characterize how the surface charge is changing as the ratios tend to increase.
Response: Following your suggestion, we have tested other ratio, and result has been added in table 1 in the revised manuscript.
- It is not necessary to include the zeta-potential chromatograms in the figure. The values should be listed in a table together with the DLS and PDI.
Response: We have listed DLS and PDI in Table 1 in the revised manuscript.
- Figure 7. The authors should study other ratios in order to evaluate the transfection efficiency.
Response: Transfection experiment have been reconducted and result have been shown in Figure 6 and Figure 7.
- The authors have used two siRNA oligonucleotides (RhoA and RhoC) however the silencing of the complexes using RNA interference is missing in this manuscript.
Response: Thank you for your suggestion, the transfection experiment showed that Arg-CS nanorods have the ability to deliver siRNA into leukemic cells (Figure 5 and Figure 6). This manuscript focus on the novel found that Arg-CS is a kind of nanomaterial that have the potential to safely deliver siRNA into leukemia cells. We focus on the siRNA delivering step, since a key challenge to realizing the broad potential of siRNA-based therapeutics is the need for safe and effective delivery methods.
- Materials and methods; lines 213-214. What does NC stand for? I think these siRNA oligonucleotides have not been used in the manuscript.
Response: NC stand for negative control, those oligonucleotides haven't been used, therefore we deleted NC in the manuscript.
(8) Line 221. Please, include the number of mmol of the reagents used for the preparation of the material
Response: The ratios were set based on reference [Biomaterials 31 (2010) 8780e8788].
(9) Line 263. The authors should indicate in the experimental part whether siRNA complexes were transfected using serum-free conditions
Response: We have described serum-free condition in the materials and method section 4.6.
Round 2
Reviewer 1 Report
No big scientific issues. But I still suggest that experiments of gene knockdown should be provided given the scope of Int. J. Mol. Sci.
Author Response
We have detected the knockdown efficiency by Western blot assay in Figure 7 in the revised manuscript.
Reviewer 2 Report
The authors have taken into consideration mostly of the suggestions by there are still some important issues with the introduction section. Neither in the introduction section nor in the discussion section, the authors have not highlighted which is the advantage of arginine. They must provide several references concerning the use of arginine in biomedical fields and to correlate the conclusions are these studies with their study.
Moreover, the following reference should be considered in the introduction section: https://doi.org/10.3390/polym13234114;
Author Response
The advantage of argnine have been used for explaining the advantage of Arg-CS in siRNA delivery in the discussion part from “The first reason for Arg-CS to compete CS in siRNA delivering may be that the Arg modification enhances the electrostatic interaction.” to “In addition, the guanidine groups of arginine also have a strong affinity to heparan sulfate, which assist the adhering to the cell membrane.”
The reference has been added in the introduction part.
Reviewer 3 Report
The authors have addressed the majority of the reviewer's comments and I recommend the publication of this manuscript in Int. J. Mol. Sci.
Author Response
Thank you for your ratification.